# Effectiveness of Implementing Hospital Wastewater Treatment Systems as a Measure to Mitigate the Microbial and Antimicrobial Burden on the Environment

**DOI:** 10.3390/antibiotics14080807

**Published:** 2025-08-07

**Authors:** Takashi Azuma, Miwa Katagiri, Takatoshi Yamamoto, Makoto Kuroda, Manabu Watanabe

**Affiliations:** 1Department of Pharmaceutical Sciences, Osaka Medical and Pharmaceutical University, Takatsuki 569-1094, Japan; takashi.azuma@ompu.ac.jp; 2Center for Infectious Disease Education and Research (CiDER), Osaka University, Suita 565-0871, Japan; 3Department of Surgery, Toho University Ohashi Medical Center, Tokyo 153-8515, Japan; katagiri@med.toho-u.ac.jp; 4Department of Medical Technology, Faculty of Health Sciences, Kumamoto Health Science University, Kumamoto 861-5598, Japan; yamataka@kumamoto-hsu.ac.jp

**Keywords:** antimicrobial resistance (AMR), hospital wastewater, ozonation, ultraviolet light-emitting diode (UV-LED), antimicrobial-resistant bacteria (ARB), metagenomics, antimicrobials

## Abstract

**Background:** The emergence and spread of antimicrobial-resistant bacteria (ARB) has become an urgent global concern as a silent pandemic. When taking measures to reduce the impact of antimicrobial resistance (AMR) on the environment, it is important to consider appropriate treatment of wastewater from medical facilities. **Methods:** In this study, a continuous-flow wastewater treatment system using ozone and ultraviolet light, which has excellent inactivation effects, was implemented in a hospital in an urban area of Japan. **Results:** The results showed that 99% (2 log_10_) of Gram-negative rods and more than 99.99% (>99.99%) of ARB comprising ESBL-producing *Enterobacterales* were reduced by ozone treatment from the first day after treatment, and ultraviolet light-emitting diode (UV-LED) irradiation after ozone treatment; UV-LED irradiation after ozonation further inactivated the bacteria to below the detection limit. Inactivation effects were maintained throughout the treatment period in this study. Metagenomic analysis showed that the removal of these microorganisms at the DNA level tended to be gradual in ozone treatment; however, the treated water after ozone/UV-LED treatment showed a 2 log_10_ (>99%) removal rate at the end of the treatment. The residual antimicrobials in the effluent were benzylpenicillin, cefpodoxime, ciprofloxacin, levofloxacin, azithromycin, clarithromycin, doxycycline, minocycline, and vancomycin, which were removed by ozone treatment on day 1. In contrast, the removal of ampicillin and cefdinir ranged from 19% to 64% even when combined with UV-LED treatment. **Conclusions:** Our findings will help to reduce the discharge of ARB and antimicrobials into rivers and maintain the safety of aquatic environments.

## 1. Introduction

The emergence and spread of bacteria resistant to antimicrobials (antimicrobial resistance (AMR)) is increasing worldwide, making it more difficult to treat and control infectious diseases. According to a 2016 WHO O’Neill report highlighting the seriousness of this problem, deaths due to antimicrobial-resistant bacteria (ARB) are expected to increase from 700,000 in 2014 to 10 million in 2050 [1]. Subsequently, a survey of 204 countries and territories found that annual global deaths from ARB will reach 1.27 million in 2019 [2], and the most recent report for 2024 indicates that annual global deaths from ARB will exceed 39 million over the next 25 years to 2050, with 169 million related deaths [3], and the threat of a silent pandemic is becoming a reality.

Recent studies have shown that ARB enter aquatic environments through wastewater systems, creating new environmental contamination problems [4,5,6]. ARB flows into the environment are not only a potential risk for the outbreak and spread of new infectious diseases, but also pose a potential health risk to humans, animals, and the environment, and are becoming an increasingly serious problem on a global scale [7,8,9,10]. In today’s globalized world, the problem of ARB has surpassed national borders. Clarifying the current situation and developing effective countermeasures to reduce environmental risks are urgent issues [11,12,13]. WHO promotes the “One Health” approach as an all-encompassing measure for humans, animals, and the environment, and encourages countries to implement it [14,15].

Several studies have reported that ARB detected in aquatic environments include those relevant to clinical practice [16,17,18,19]. In Japan, the *bla*_NDM-5_ gene among carbapenem-resistant *Enterobacterales* (CRE), which requires clinical attention, the *bla*_CTX-M-55_ gene encoding extended-spectrum *β*-lactamase, and the *bla*_NDM-5_ gene encoding NDM-5 have been reported to be clinically relevant. *Klebsiella pneumoniae* carrying the *bla*_KPC-2_ gene for CRE and *Escherichia coli* (*E. coli*) carrying the *bla*_CTX-M-55_ gene have been detected in central Tokyo Bay [20,21,22].

The environmental impact of hospital wastewater associated with medical care is not insignificant [23,24,25]. On the other hand, although hospital wastewater regulations have established standards for wastewater discharge to the sewer system based on basic water quality elements for general business establishments, new environmental pollutants such as ARB and antimicrobials are still under development and have only recently begun to be recognized and discussed [26,27,28]. Given this background, taking measures to treat wastewater properly before it is discharged from hospitals into sewer systems is an effective way to address the problem of ARB. In fact, studies have shown that in the case of pharmaceuticals, the percentage of environmental impact from hospital wastewater can range from tens of percent to 71% [29,30,31,32]. However, despite the importance of hospital wastewater, there is limited knowledge regarding the status of ARB and antimicrobials in hospital wastewater worldwide and their treatment methods, and there are still issues to be addressed to solve the problem of ARB from an environmental perspective [33,34,35]. With the recent remarkable development of science and technology, advanced wastewater treatment systems have been developed that are also effective in treating hospital wastewater including the use of Fenton [36,37], electrolysis [38], TiO_2_ [39], persulfate [40], UV/chlorine [41], and ozone [37,42]. Among them, ozone treatment has been the focus of much research in recent years because it has strong potential in terms of both sterilizing and removing pollutants, including deodorizing without the addition of any chemicals, and the wastewater is free from residues after treatment [43,44,45]. However, the efficacy of wastewater systems based on ozone treatment for hospital wastewater has been primarily evaluated in small-scale (several hundreds of mL to several L) test systems in laboratories [46,47], and much has yet to be investigated on the actual hospital wastewater scale from a global point of view [48,49].

Under these social circumstances, our research group has recently discussed the ideal treatment of hospital wastewater and has pioneered the social implementation of an advanced wastewater treatment system using ozone, which has excellent effects in both water purification and wastewater treatment, and has been introduced in some water and wastewater treatment plants in Japan [50,51]. As a first step, a closed system using a storage-type batch treatment tank was used to treat a portion (1 m^3^) of the hospital wastewater [52], and as a next step, a continuous-flow treatment system was used to treat all of the wastewater flowing from the hospital to the public sewer system [53].

The results of these studies have shown that in a closed-batch treatment system with a constant volume of treated water, both ARB and antimicrobials are reduced to the lower detection limit after 20 min of ozone treatment, which is comparable to the inactivation rate (approximately 99% or higher [54,55]) in wastewater treatment plants using ozone treatment [52]. However, flow-through treatment has been shown to be more effective than ozonation in hospitals. However, when all hospital wastewater is treated in a continuous-flow system, some ARB and antimicrobials tend to remain in the treated water, making it difficult to achieve sufficient inactivation at an inactivation rate estimated at 90% or less [53]. The challenge was to design a treatment system with high performance and efficiency to treat large volumes of wastewater and to achieve highly effective inactivation.

In this study, we aimed to construct an effective treatment system for hospital wastewater, and we improved the efficiency of ozone treatment by returning to the reaction tank all the excess ozone exhaust gas that is not used in the reaction but is transferred from the hospital wastewater to the gas phase and also expanded and optimized the functions of the treatment tank to enable the inhibition of biofilm formation in the tank, which would reduce the inactivation effect. In addition, the system was verified as an advanced wastewater treatment system that combines treatment with additional ultraviolet light-emitting diode (UV-LED) irradiation of the treated water after ozonation and its effectiveness as a measure to reduce the impact of ARB and antimicrobials on the environment. The effectiveness of the system in terms of reducing the impact of ARB and antimicrobials on the environment was evaluated.

## 2. Materials and Methods

### 2.1. Hospital Wastewater

Ozone-based microbial and antimicrobial inactivation studies were conducted at the University Hospital, Ohashi Medical Center (BN; 35.652578° N, 139.683959° E), with a capacity of 319 beds, located in Tokyo, Japan, at Toho University, as previously reported [53]. At this hospital, wastewater was collected in two underground storage tanks (original storage tank (influent)) with a volume of 22.5 m^3^ and discharged into the public sewage system at a frequency of 2 m^3^ per discharge or approximately 25 times per day. The basic water quality parameters of the hospital wastewater prior to this study were as follows: chemical oxygen demand (COD), biochemical oxygen demand (BOD), suspended solids (SSs), and total nitrogen (TN) were 254 mg/L, 112 mg/L, 87 mg/L, and 82 mg/L, respectively.

### 2.2. Hospital Wastewater Treatment by Ozone and UV with Continuous-Flow System

An overview and configuration of the continuous ozone and UV hospital wastewater system implemented in this study are shown in Figure 1. Untreated hospital wastewater is introduced into a cylindrical ozonation tank (wastewater treatment tank 1) with a volume of 0.68 m^3^ using a submersible pump (40DWV5.15A, Ebara Corporation, Tokyo, Japan) at a flow rate of 20 L/min. In the ozonization tank, an ozone generator (Ozonia^®^ TOGC45X, Veolia Environnement S.A., Paris, France) was used to generate ozone gas from the oxygen in the air at a flow rate of 5 L/min (equivalent to an ozone generation rate of 42 g/h and an effective ozone gas concentration of 139 mg/L). A submersible pump (50DB) equipped with fine bubble-generating nozzle (YJ-11, For EARTH Co., Ltd., Tokyo, Japan) was used to introduce the ozone gas generated from oxygen into the air at a flow rate of 5 L/min (equivalent to an ozone generation rate of 42 g/h and an effective ozone gas concentration of 139 mg/L). A submersible pump (50DWV5.75B, Ebara Corporation, Tokyo, Japan) with microbubbles with diameters between 1 μm and 100 μm in the 30–50 μm mode and ultrafine bubbles with diameters less than 1 μm in the 50–200 nm range [56,57] were used to fill the ozone treatment tank. In this study, excess ozone gas that does not dissolve in wastewater and migrates to the gas phase as flue gas, was reintroduced into the submersible pump to increase the efficiency of the amount of ozone used in the treatment. In addition, the ozonation tank contains 30 10 cm diameter spherical polypropylene skeleton balls (BS-100-N, Wuxi BioSource Environmental Equipment Co., Wuxi, China) with weights, which, together with the cylindrical ozonation tank, agitated the treated water and prevented biofilm formation on the tank walls (Appendix A). The tests were conducted at an average dissolved ozone concentration of 3.4 mg/L, which was approximately three times higher than the 1 mg/L reported in a previous study [53].

After ozonation, the wastewater is transferred at a flow rate of 20 L/min by overflow through a square stainless steel screen layer (0.12 m^3^) with one 10 mm mesh, one 5 mm mesh, and one 3 mm mesh, arranged in steps of increasing fineness to remove suspended solids, subsequently into a rectangular UV treatment tank (wastewater treatment tank 2). The wastewater was transferred from the UV treatment tank to a UV treatment unit (UV-LED system (DWM13-S06-XX-K, NIKKISO Co., Ltd., Tokyo, Japan) with a peak emission of 280 nm and an intensity of 43 mW/cm^2^). The UV-treated water was then returned to the original storage tank using a submersible pump (50DWV5.4B, Ebara Corporation, Tokyo, Japan) at a flow rate of 185 L/min. A 100 mL aliquot of the solution was collected 0, 1, 2, 3, 4, and 5 days after the start of the experiment from each tank daily.

### 2.3. Viable Bacterial Counting of Wastewater Samples

Basically, most methods were previously described in Azuma et al. [53]. Briefly, an aliquot (100 µL) of influent or treated wastewater sample was 10-fold serially diluted with phosphate-buffered saline and then spread on non-selective media BTB (Bromothymol Blue, lactose agar; Drigalski Agar, Modified) agar and CHROMagar ESBL plates (bioMérieux S.A., Marcy-l’Étoile, France) for extended-spectrum *β*-lactamase (ESBL)-producing bacteria. Colony-forming units per mL (CFU/mL) were determined at the appropriate dilution for each treatment time point.

### 2.4. Metagenomic DNA-Seq Analysis of Wastewater Samples

Most methods have been previously described [53]. Bacteria were collected by passing through TPP Rapid Filtermax Vacuum Filtration systems (TPP, Trasadingen, Switzerland) in a 100 mL bottle. One-fourth of the collected membrane, corresponding to 25 mL of influent or treated water, was cut into small pieces, followed by bead-beating in ZR-96 BashingBead Lysis Tubes (0.1 mm and 0.5 mm; Zymo Inc., Irvine, CA, USA) using a GenoGrinder 2010 homogenizer (SPEX SamplePrep, Metuchen, NJ, USA). After brief centrifugation, the supernatant was purified using a QIAGEN minielute PCR purification kit. Metagenomic DNA-seq libraries were prepared using the QIAseq FX DNA Library Kit (Qiagen, Hilden, Germany), followed by short-read sequencing using the NextSeq 500 platform (2 × 70-mer paired-end) (Illumina, San Diego, CA, USA). Adapter and low-quality sequences were trimmed using the fastp program, and the taxonomic classification of every read from the metagenomic analysis was performed using CZ ID website analysis (Minimap2 and Diamond read alignment to identify the taxonomy).

Resistome analysis using metagenomic DNA-Seq reads was performed using ARGs-OAP v3.2.1 against the implemented antibiotic resistance gene (ARG) database [58,59].

All raw read sequence files were deposited into and are available from the DRA/SRA database (Appendix A).

### 2.5. Antimicrobial Analysis in Hospital Wastewater Samples

A total of 17 antimicrobials were evaluated in five categories, as shown in Appendix A. The categories analyzed were *β*-lactams (ampicillin, benzylpenicillin, cefdinir, cefpodoxime, cefpodoxime proxetil, and ceftiofur), new quinolones (ciprofloxacin, enrofloxacin, and levofloxacin), macrolides (azithromycin and clarithromycin), tetracyclines (chlortetracycline, doxycycline, minocycline, oxytetracycline, and tetracycline), and glycopeptides (vancomycin). This selection was based on previous reports on the occurrence and frequency of detection in hospital wastewater, WWTPs, and river water, both in Japan and worldwide [53,60,61], as well as on antimicrobial use in medical settings in Japan [62,63]. All analytical standards were of high purity (>98% purity).

The analysis of target antimicrobials in hospital wastewater was conducted using a combination of SPE and ultra-performance liquid chromatography–tandem mass spectrometry (UPLC–MS/MS), as previously described [64,65]. Briefly, 10 mL of each wastewater sample was filtered through the TPP Rapid Filtermax Vacuum Filtration systems, as described in Section 2.4., and the filtrate was passed through the OASIS HLB syringe barrel cartridges with a 200 mg solid-phase carrier and 6 mL column (Waters Corp., Milford, MA, USA) at a flow rate of 1 mL/min. All cartridges were cleaned by washing with 6 mL Milli-Q water (Millipore, Burlington, MA, USA), pre-adjusted to pH 3, and dried using a vacuum pump. The adsorbed antimicrobials were eluted with 3 mL acetone and 3 mL methanol, and then evaporated mildly to dryness under a gentle stream of N_2_ gas at 37 °C. The residue was solubilized in 200 μL of a 90:10 (*v*/*v*) mixture of 0.1% formic acid solution in methanol, and 10 μL of this solution was analyzed using a UPLC–MS/MS fitted with a column (2.1 mm × 100 mm, 1.7 μm) UPLC BEH C_18_ (Waters Corp.) coupled to a tandem quadrupole mass spectrometer (TQD, Waters Corp.). The measurement conditions for the LC-MS/MS measurements of each antibacterials are shown in Appendix A.

The quantification involved subtracting the blank data from the corresponding data obtained from the spiked sample solutions to account for matrix effects and losses during sample extraction [66,67]. The recovery rates were calculated from the deviations between the spiked and standard calibration data [68,69]. The validation of the recovery rates of each antimicrobial in the wastewater ranged from 58% to 122% (Appendix A). The limits of detection (LODs) and limits of quantification (LOQs) were calculated as the concentrations at signal-to-noise ratios of 3 and 10 [70,71], and are summarized in Appendix A. These profiles were generally similar to those previously reported for pharmaceuticals in river water and wastewater samples [68,72,73].

## 3. Results

### 3.1. Proportion of Bacteria in Hospital Wastewater

Hospital wastewater was treated with ozone followed by UV-LED irradiation in a continuous-flow pilot plant (Figure 1) for 0, 1, 2, 3, 4, and 5 days after starting continuous treatment. The visible brown color of the wastewater disappeared on day 1, and continuous treatment kept the treated water clear during treatment (Figure 2A).

The treated water collected from the ozone or UV-LED tanks showed a 2 log_10_ (>99%) inactivation of viable bacteria on BTB agar after 1 day of treatment (Figure 2B,C), and it maintained the level of reduced viable bacteria throughout the treatment period (Figure 2C).

Compared with the bacterial CFU on BTB agar, the potential ESBL-producing *Enterobacterales* on CHROMagar ESBL demonstrated that ozone showed remarkably effective inactivation of almost 2 log_10_ (>99%) (Figure 2D). Notably, UV-LED treatment showed incredible inactivation of the bacteria, which finally underwent 4 log_10_ (>99.99%) inactivation through day 1 to day 5 (Figure 2D).

To characterize the actual susceptibility of bacterial genera to ozone and UV, membrane-trapped bacteria in the treated sample (corresponding to the 25 mL water sample) were subjected to genomic DNA extraction, and metagenomic DNA-Seq analysis was performed (summarized in Appendix A).

Although viable bacteria decreased significantly on day 1 after treatment (Figure 2C), metagenomic DNA-Seq showed that removal at the DNA level for these microorganisms tended to be gradual. Interestingly, for *Bacteroides*, *Prevotella*, *Escherichia*, *Klebsiella*, *Acinetobacter*, and *Pseudomonas*, the treated water after ozone treatment contained approximately the same number of gene copies as the influent water and showed no clear removal effect. However, the treated water after ozone/UV-LED treatment showed an average inactivation of about 50% for all DNAs, and the removal rate reached 2 log_10_ (>99%) at the end of the treatment test (day 5) (Figure 3).

### 3.2. Resistome Analysis in Hospital Wastewater

In addition to the bacterial taxonomic analysis, ARG resistome analysis was performed using metagenomic DNA-Seq reads. The top 12 most abundant ARGs are summarized in Table 1 (all results obtained for other ARGs are summarized in Appendix A). Class 1 integrons (*sul1* and *qacEdelta*), *β*-lactamase GES variants (*bla*_GES-15_, *bla*_GES-20_, and *bla*_OXA-1_), aminoglycoside acetyl transferase (*aac(6′)*-*31*), and tetracycline resistance (*tet*(Q) and *tet*(36)) were mainly detected in sewage influent samples. Most of these were significantly inactivated on days 3 or 4 post treatment, consistent with the results of the metagenomic DNA-Seq data (Figure 3). However, *sul1* and *aac(6′)*-*31* increased again during the experiment (Table 1).

### 3.3. Removal of Antimicrobials in Hospital Wastewater

Eleven antimicrobials were detected in untreated hospital wastewater. The classification and detected concentrations of each antimicrobial were *β*-lactams (157 μg/L for ampicillin, 1.5 μg/L for benzylpenicillin, 21 μg/L for cefdinir, and 466 ng/L for cefpodoxime), new quinolones (57 ng/L for ciprofloxacin and 19 μg/L for levofloxacin), macrolides (1.2 μg/L for azithromycin and 676 ng/L for clarithromycin), tetracyclines (101 ng/L for doxycycline and 1.1 μg/L for minocycline), and glycopeptides (27 μg/L for vancomycin). The higher concentrations of antimicrobials were consistent with those previously reported for hospital wastewater. The time course of the antimicrobial concentrations in hospital wastewater during ozone treatment followed by UV treatment is summarized in Table 2 and Figure 4.

Ozonation rapidly removes antimicrobials from wastewater, with benzylpenicillin, cefpodoxime, ciprofloxacin, levofloxacin, azithromycin, clarithromycin, doxycycline, minocycline, and vancomycin being reduced to below the detection limit with removal rates generally maintained at 99% to 99.99% (2 log_10_ to 4 log_10_ removal) within 1 to 2 days of treatment. This trend was similar throughout the treatment period, with only levofloxacin occasionally detected during the treatment period; however, a high removal rate was maintained, with an average detected concentration of 363 ng/L ± 406 ng/L (corresponding to a removal rate of 98% ± 2%). On the other hand, ampicillin and cefdinir were detected at 189 μg/L ± 12 μg/L and 7.6 μg/L ± 2.3 μg/L, respectively (removal rates of 19% ± 24% and 64% ± 11%) throughout the treatment period, although the concentration of antimicrobials was lower than in the untreated effluent, indicating that the ozone treatment was effective. The concentrations of these antimicrobials in the UV-LED treatment tank were 96 µg/L ± 30 µg/L and 6.8 g/L ± 1.5 µg/L, and the removal rates of these antimicrobials were 41% ± 34% and 13% ± 14% (in terms of cumulative removal over the entire treatment period, 39% ± 19% and 80% ± 45%) for the effluent of the wastewater treatment tank 1 (ozone), respectively.

## 4. Discussion

The present investigation has demonstrated that the entire volume of wastewater actually generated in hospitals can be continuously treated by an advanced flow-through continuous wastewater treatment system to inactivate ARB and residual antimicrobials entering the public sewer system on-site and in real time. The results of this study will contribute to the future of hospital wastewater treatment and the promotion of measures to reduce the impact of AMR on the environment and humans through the social implementation of hospital wastewater treatment systems on a regional and larger scale.

Ozone treatment provided the most effective treatment for the microorganisms present in hospital wastewater, while ozone followed by UV-LED treatment provided the most complete removal, including genetic material. Viable microorganisms detected on the culture medium are considered to be those that can directly affect human and animal health through infection [74,75]. On the other hand, previous studies have shown that the direct health effects of microbial genes through ingestion are considered low, but may pose an indirect risk of generating new ARB through horizontal transmission or transformation [76,77]. Here, although viable bacterial counts decreased significantly on day 1 post treatment (Figure 2C), metagenomic DNA sequencing revealed that the reduction in microbial DNA occurred more gradually. This suggests that bacterial viability is rapidly lost upon initial contact with ozone molecules, but fragments of genomic DNA may persist in the solution, indicating that additional time is required for complete DNA degradation.

Class 1 integrons detected in hospital wastewater are involved in the horizontal spread of ARGs by incorporating them into gene cassettes and may serve as indicators when considering the impact of ARB on the environment [78,79]. The fact that the various ARGs in this study can all be reduced below the detection limit by advanced treatment is an important finding when considering the appropriate treatment of hospital wastewater for environmental impact.

For the residual antimicrobials in hospital wastewater, it was assumed that the majority of the treatment effect would be achieved by ozone treatment. This result was based on the fact that the contact time with UV in the UV treatment tank is a few seconds, and that the inactivation effect of UV is expected to progress as the substance to be inactivated is irradiated with UV [80,81], and the effect is exerted when the wavelength of the irradiated UV and the absorption wavelength of the substance to be inactivated approximately match [82,83], which was generally a reasonable result. The amount of residual antimicrobials in the wastewater increases or decreases due to the influence of untreated wastewater continuously flowing from the original storage tank, but the ozone treatment and UV-LED treatment maintain the inactivated state of the agent.

Previous studies have demonstrated that ozonation and/or UV treatment generally reduce the ecotoxicity of wastewater effluents compared to untreated counterparts [44,84,85]. However, in certain instances, these advanced oxidation processes may paradoxically increase toxicity, likely due to the formation of reactive transformation products [85]. The strong oxidative capacity of ozone and hydroxyl radicals can minimize the generation of such undesirable transformation products when sufficient processing times are applied or when used in conjunction with catalysts, such as UV irradiation and hydrogen peroxide, thereby enhancing the degradation of persistent organic pollutants [43,86,87]. To comprehensively evaluate these complex interactions, the implementation of whole-effluent toxicity (WET) testing, as recommended by the U.S. Environmental Protection Agency, provides an integrative assessment of toxicological impacts across multiple trophic levels and can serve as a robust framework for assessing the efficacy and safety of advanced treatment processes in wastewater management [88,89].

The cost of the system is approximately USD 74,000 for the ozone-only wastewater treatment system as a package in 2025, and USD 105,000 for ozone and UV-LED wastewater treatment systems in 2025. In the case of a wastewater treatment system using ozone and UV-LED, the cost would be USD 105,000. The cost of a complete set of ozone treatment equipment is USD 29,000, and that of a complete set of UV-LED treatment equipment is USD 8600. As ozone generation equipment becomes more powerful and energy-efficient with the development and advancement of science and technology, it is expected to become more popular at the same price and performance. Consequently, the use of ozone generators with similar performance and efficiency at similar prices is expected to become more widespread [90,91,92]. The maintenance cost was estimated to be approximately USD 1060, including the replacement of ozone generator parts, replacement of submersible pump packing, and periodic inspections. The power consumption required to operate this hospital wastewater treatment system continuously for one day is 127 kWh, or approximately USD 23, including 50.4 kWh USD 9 for the pump, 74.4 kWh USD 13 for the ozone treatment unit, and 2.2 kWh USD 0.4 for the UV-LED treatment equipment. Considering that the daily power consumption of an average household in Japan is 11.4 kWh USD 2, this power consumption is not low, but is unlikely to be a major problem preventing the introduction of these systems in a typical office with a large number of employees [93]. It should also be noted that power consumption can be achieved using 10 high-efficiency solar panels, each of which generates 9.6 kWh of electricity per day [94,95]. This is a significant advantage for the cost-effective inactivation of hospital wastewater. These results suggest that the introduction of advanced hospital wastewater treatment systems may be an effective approach to reduce the burden of AMR in the environment.

This study presents an advanced hospital wastewater treatment system that demonstrates high efficacy in mitigating the spread of AMR. The system offers enhanced practicality for real-world application by preventing biofilm formation, even under continuous operation. Notably, this is the first study to construct and evaluate a system capable of treating the entire volume of hospital wastewater discharged into the public sewage system while maintaining stable performance under continuous use. This system provides a new perspective and a potential solution for controlling the dissemination of AMR within the environment while establishing an environmentally sustainable approach to hospital wastewater management. Attempts to gather and review scientific evidence aimed at realizing a society that can enjoy safety in response to health risks, including ARB, by taking a comprehensive view of medical effluents circulating in the environment with people and society suggest the importance of considering the relationship between medicine and the environment, which will become increasingly important in the future to achieve both modern and affluent lifestyles and sustainable development and prosperity for humankind. It is expected to become increasingly important in the future to achieve both modern and affluent lifestyles, as well as sustainable development and human prosperity [12,61,76].

Global discussions are currently underway regarding the appropriate management and discharge of hospital wastewater to mitigate its environmental and public health impacts [96,97,98]. In Japan, the national Action Plan also emphasizes addressing AMR in the environment as a critical component of AMR countermeasures [99,100]. The findings of this study offer valuable insights into the effective treatment of hospital wastewater, contributing to both environmental protection and the safeguarding of public health. The results of this study have been implemented in society at an early stage in hospital facilities, which are pioneers in this field, and will contribute to the study of similar countermeasures not only in Japan, but also in developed countries in Europe, the United States, and other rapidly developing countries in Asia and Africa. In the future, it is important to consider how to utilize the results of this research, for example, through technology transfer in cooperation with domestic medical institutions and local governments, the assessment and management of health risks associated with hospital wastewater, and the development of guidelines for risk control methods [27,96,101,102,103]. In addition, it is important to promote further research and development that reflects social conditions through the exchange of opinions with related ministries and agencies.

## 5. Limitations

An ideal hospital wastewater treatment system should discharge treated water directly into the public sewer system. Further improvements to these aspects of treatment equipment should be investigated in the future.

The treatment effectiveness for the basic general water quality parameters was also noted. For hospital wastewater treatment in this study, the treatment system was found to be capable of inactivating ARB and antimicrobials, which have become new environmental pollutants of concern in recent years, although some improvements, including in water quality, have not yet been achieved. Improving the treatment to include these water quality items can be achieved by increasing the amount of ozone injected; however, in some respects, this treatment strategy is not practical or energy-efficient, and it would be more effective in combination with other treatments. These issues can be resolved via biodegradation, as ozone treatment alone makes it difficult to convert persistent substances into biodegradable substances. Our results support the need for further conclusive research that considers experimental, technical, and regional customs, bias, and unknown factors.

## 6. Conclusions

We developed an advanced treatment system that can continuously treat all wastewater discharged from hospitals to public sewer systems on-site and evaluated its effectiveness in terms of inactivating ARB and residual antimicrobials inherent in wastewater as emerging contaminants that may pose a risk to the environment and human health. Several ARB, clinically important class 1 integrons, and residual antimicrobials were detected in untreated hospital wastewater, most of which were inactivated by ozone treatment. Furthermore, metagenomic analysis revealed that ozone treatment followed by ultraviolet treatment using UV-LEDs resulted in >99% (2 log_10_) ARG removal. These inactivation effects were maintained throughout the treatment trials in this study, and the overall results facilitate a comprehensive understanding of the AMR risk posed by hospital wastewater, and provide insights for devising strategies to eliminate or mitigate the environmental impact of ARB and residual antimicrobials. It is important to promote the required level of reduction and further study the practical aspects of implementing such technologies, as well as the packaging of equipment systems. Additionally, it is important to deepen social understanding and support hospital incentives. Our findings contribute to a better understanding of the environmental management of hospital wastewater and enhance the effectiveness of on-site disinfection wastewater treatment systems at medical facilities for mitigating the discharge of pollutants into aquatic environments. Moreover, our results will help to reduce the discharge of ARB and antimicrobials into rivers and maintain the safety of aquatic environments.

## Figures and Tables

**Figure 1 antibiotics-14-00807-f001:**
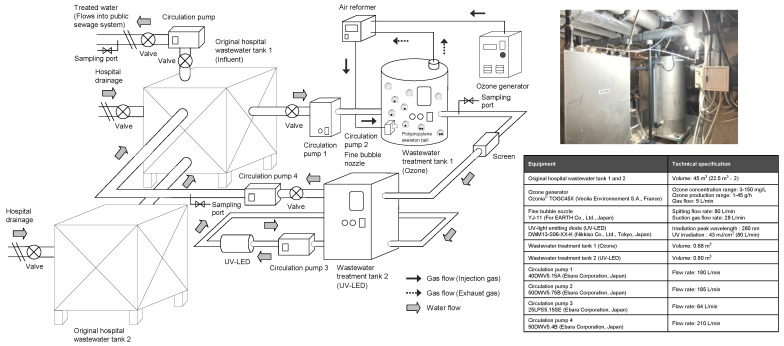
Schematic representation of continuous-flow ozone treatment followed by an ultraviolet light-emitting diode (UV-LED) system implemented at a hospital facility. The image depicts the appearance of the hospital wastewater disinfection treatment system. The technical specifications of the equipment used in the system are described in the figure above.

**Figure 2 antibiotics-14-00807-f002:**
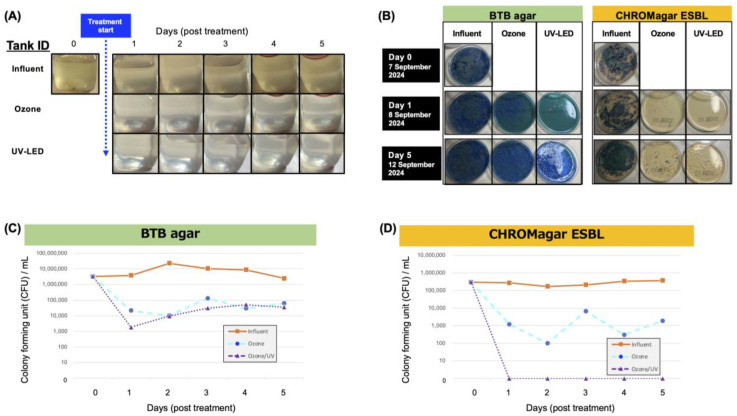
Characterization of treated wastewater by bacterial viability. (**A**) Visual image of the collected waste or treated wastewater after ozone or UV-LED treatment. (**B**) Isolation of bacteria from treated wastewater samples on BTB agar and CHROMagar ESBL. An aliquot (100 µL) of influent, ozone-, or UV-treated wastewater samples was spread on an agar plate at 10-fold dilution. (**C**,**D**) Colony forming units per milliliter (CFU/mL) were determined at the appropriate dilution for each treatment time point.

**Figure 3 antibiotics-14-00807-f003:**
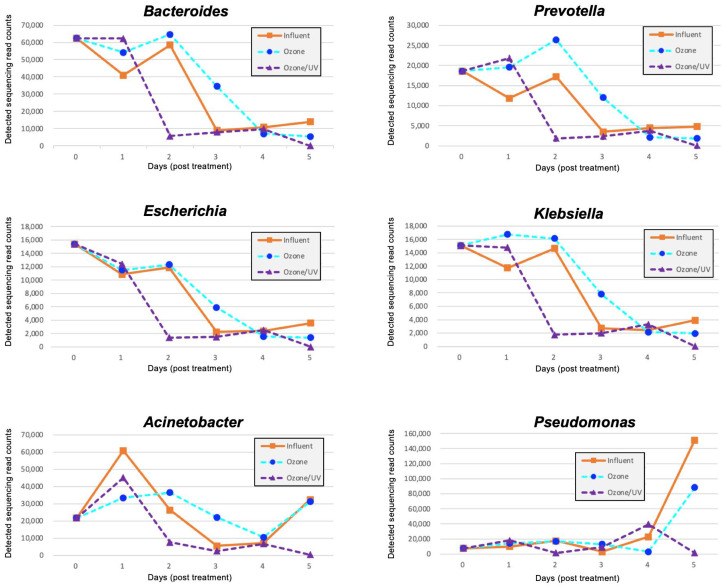
Metagenomic DNA-seq analysis of bacteria trapped on a 0.2 µm filter after ozone–UV-LED treatment. Notable bacterial genera (*Bacteroides*, *Prevotella*, *Escherichia*, *Klebsiella*, *Acinetobacter*, and *Pseudomonas*) were highlighted to show the metagenomic sequencing read counts detected by megablast search and subsequent taxonomic classification using the CZ ID online tool. The results for each bacterial genus are summarized in Appendix A.

**Figure 4 antibiotics-14-00807-f004:**
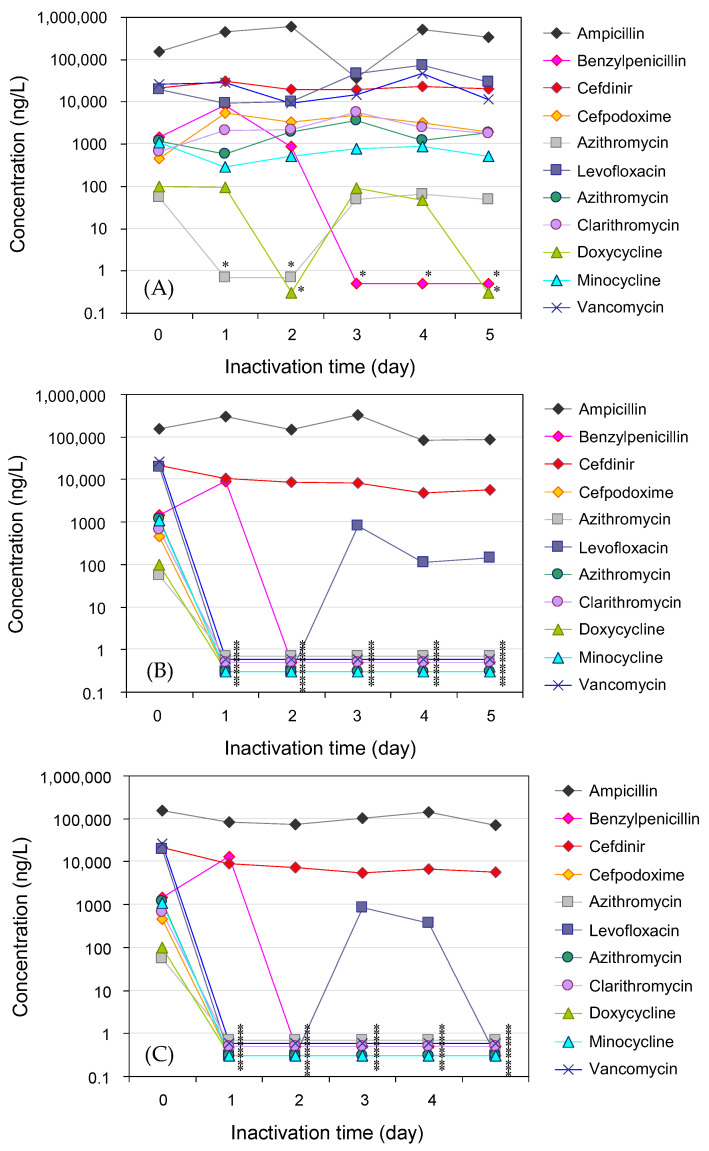
Time course of antimicrobial concentrations in hospital wastewater during ozone treatment. Removal of antimicrobials over time during the treatment of hospital wastewater. ((**A**) Influent (original storage tank), (**B**) ozone treatment (wastewater treatment tank 1), and (**C**) UV-LED treatment (wastewater treatment tank 2)). The value is below the lower limit of detection, as indicated by the * symbol.

**Table 1 antibiotics-14-00807-t001:** Metagenomic DNA-Seq analysis of antimicrobial resistance genes (ARGs) after ozone–UV-LED treatment.

**Tank ID**	Days (Post Treatment)	Total ARGs	Detected RPM (Reads Per Million) of Antimicrobial Resistance Genes (ARGs) by ARGs_OAP Search *											
			Sulfonamide *sul1*	Sulfonamide *sul2*	Multidrug *qacEdelta1*	*β*-Lactam GES-15	*β*-Lactam GES-20	*β*-Lactam OXA-1	Aminoglycoside *AAC(6′)*-*31*	Aminoglycoside *aadS*	Aminoglycoside *APH(3″)*-*Ib*	Aminoglycoside *APH(6)*-*Id*	**Tetracycline *tet*(36)**	**Tetracycline *tet*(Q)**
Original storage tank (Influent)	0	2598	315	42	181	228	41	137	130	72	40	65	70	60
	1	2527	396	32	76	137	15	80	216	76	76	0	103	34
	2	2723	333	52	187	237	44	81	99	12	24	6	53	42
	3	2241	0	0	0	0	0	0	0	0	379	0	158	158
	4	811	281	0	0	0	0	0	470	0	0	0	0	0
	5	1066	471	0	0	0	0	0	234	0	0	0	0	0
Wastewater treatment tank 1 (ozone)	0	2598	315	42	181	228	41	137	130	72	40	65	70	60
	1	2701	340	13	62	124	25	331	99	12	75	129	50	50
	2	2821	480	16	263	226	30	103	47	60	69	33	67	40
	3	3529	803	0	275	221	0	311	172	0	313	114	99	49
	4	6330	944	0	2980	398	0	0	0	0	428	0	0	0
	5	1561	0	0	0	0	0	0	0	0	0	0	0	0
Wastewater treatment tank 2 (UV-LED)	0	2598	315	42	181	228	41	137	130	72	40	65	70	60
	1	2270	196	0	70	310	0	136	89	14	65	44	32	82
	2	2075	1754	0	0	0	0	0	0	0	0	0	0	0
	3	3055	554	0	0	1077	0	1345	0	0	0	0	0	0
	4	760	0	321	0	0	0	0	0	0	0	0	0	0
	5	0	0	0	0	0	0	0	0	0	0	0	0	0

* Notable top 12 ARGs were selected to show the sequencing reads corresponding to the targeted ARGs by the ARGs_OAP program. DNA conc. (ng/µL) and Metagenomic DNA-Seq (total reads) are shown in Appendix A, and all obtained results for other ARGs are also summarized in Appendix A.

**Table 2 antibiotics-14-00807-t002:** A summary of antimicrobial concentrations in hospital wastewater during continuous-flow treatment (N.D.: not detected) ((A) influent (original storage tank), (B) ozone treatment (wastewater treatment tank 1), and (C) UV-LED treatment (wastewater treatment tank 2)).

Classification	Antimicrobials	Inactivation Time (day)					
		0	1	2	3	4	5
**(** **A)**							
*β*-lactams	Ampicillin	157,128	462,024	614,340	37,108	525,180	337,424
	Benzylpenicillin	1454	8402	871	N.D.	N.D.	N.D.
	Cefdinir	21,138	30,469	19,496	19,659	23,451	20,793
	Cefpodoxime	466	5551	3293	4865	3223	1929
	Cefpodoxime proxetil	N.D.	N.D.	N.D.	N.D.	N.D.	N.D.
	Ceftiofur	N.D.	N.D.	N.D.	N.D.	N.D.	N.D.
New quinolones	Ciprofloxacin	57	N.D.	N.D.	49	65	49
	Enrofloxacin	N.D.	N.D.	N.D.	N.D.	N.D.	N.D.
	Levofloxacin	19,404	9250	10,096	47,668	74,311	30,386
Macrolides	Azithromycin	1178	579	1925	3575	1245	1871
	Clarithromycin	676	2087	2190	5689	2453	1792
Tetracyclines	Chlortetracycline	N.D.	N.D.	N.D.	N.D.	N.D.	N.D.
	Doxycycline	101	96	N.D.	92	48	N.D.
	Minocycline	1077	293	522	793	893	516
	Oxytetracycline	N.D.	N.D.	N.D.	N.D.	N.D.	N.D.
	Tetracycline	N.D.	N.D.	N.D.	N.D.	N.D.	N.D.
Glycopeptides	Vancomycin	26,587	28,503	9563	14,733	46,503	11,753
**(B)**							
*β*-lactams	Ampicillin	157,128	297,428	148,632	329,932	84,192	87,156
	Benzylpenicillin	1454	9098	N.D.	N.D.	N.D.	N.D.
	Cefdinir	21,138	10,644	8701	8234	4874	5662
	Cefpodoxime	466	N.D.	N.D.	N.D.	N.D.	N.D.
	Cefpodoxime proxetil	N.D.	N.D.	N.D.	N.D.	N.D.	N.D.
	Ceftiofur	N.D.	N.D.	N.D.	N.D.	N.D.	N.D.
New quinolones	Ciprofloxacin	57	N.D.	N.D.	N.D.	N.D.	N.D.
	Enrofloxacin	N.D.	N.D.	N.D.	N.D.	N.D.	N.D.
	Levofloxacin	19,404	N.D.	N.D.	832	114	144
Macrolides	Azithromycin	1178	N.D.	N.D.	N.D.	N.D.	N.D.
	Clarithromycin	676	N.D.	N.D.	N.D.	N.D.	N.D.
Tetracyclines	Chlortetracycline	N.D.	N.D.	N.D.	N.D.	N.D.	N.D.
	Doxycycline	101	N.D.	N.D.	N.D.	N.D.	N.D.
	Minocycline	1077	N.D.	N.D.	N.D.	N.D.	N.D.
	Oxytetracycline	N.D.	N.D.	N.D.	N.D.	N.D.	N.D.
	Tetracycline	N.D.	N.D.	N.D.	N.D.	N.D.	N.D.
Glycopeptides	Vancomycin	26,587	N.D.	N.D.	N.D.	N.D.	N.D.
**(C)**							
*β*-lactams	Ampicillin	157,128	85,295	75,486	104,616	144,181	71,242
	Benzylpenicillin	1454	12,853	N.D.	N.D.	N.D.	N.D.
	Cefdinir	21,138	9179	7220	5449	6747	5592
	Cefpodoxime	466	N.D.	N.D.	N.D.	N.D.	N.D.
	Cefpodoxime proxetil	N.D.	N.D.	N.D.	N.D.	N.D.	N.D.
	Ceftiofur	N.D.	N.D.	N.D.	N.D.	N.D.	N.D.
New quinolones	Ciprofloxacin	57	N.D.	N.D.	N.D.	N.D.	N.D.
	Enrofloxacin	N.D.	N.D.	N.D.	N.D.	N.D.	N.D.
	Levofloxacin	19,404	N.D.	N.D.	865	371	N.D.
Macrolides	Azithromycin	1178	N.D.	N.D.	N.D.	N.D.	N.D.
	Clarithromycin	676	N.D.	N.D.	N.D.	N.D.	N.D.
Tetracyclines	Chlortetracycline	N.D.	N.D.	N.D.	N.D.	N.D.	N.D.
	Doxycycline	101	N.D.	N.D.	N.D.	N.D.	N.D.
	Minocycline	1077	N.D.	N.D.	N.D.	N.D.	N.D.
	Oxytetracycline	N.D.	N.D.	N.D.	N.D.	N.D.	N.D.
	Tetracycline	N.D.	N.D.	N.D.	N.D.	N.D.	N.D.
Glycopeptides	Vancomycin	26,587	N.D.	N.D.	N.D.	N.D.	N.D.

## Data Availability

All raw read sequence files are available from the DRA/SRA database (accession numbers DRR657618 –DRR657633 [see Appendix A]).

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
