# Peer review of "Effectiveness of Implementing Hospital Wastewater Treatment Systems as a Measure to Mitigate the Microbial and Antimicrobial Burden on the Environment"

_antibiotics, 2025, doi:10.3390/antibiotics14080807_

Round 1

Reviewer 1 Report

Comments and Suggestions for Authors

I congratulate you on your tangible and applied research. Although much research has been done on antibiotic disposal, this case highlights the application in a closed system with costing. Your research 10.3390/antibiotics12050932 is very similar to this one with the exception of the number of antibiotics studied. That is why you should focus on what is the difference and the importance of analyzing the 11 antibiotics included here, within the same category.

I would like to make a few comments for improvement. 

Sometimes the information is repeated and can become confusing (144-145 and 153). Based on this, the writing needs to be improved. 
In methodology it is mentioned that a screen is used, but no specifications are given (155).
It is not clear how many samples were taken, and if it is only one, perhaps it is not significant in the day to day life of a hospital. Were repetitions made; this is due to the complexity of the sample being real wastewater. 
Also the figures and graphics need to improve their quality. And although images help to see the big picture, graphs are more objective (Figure 2, 3), so only quality graphs are sufficient. 
It would be interesting to do a statistical study to affirm that ozone really makes a big difference with respect to the reference. In figure 3 it seems that these two have the same behavior. And this comparison does not come in the paper. 
The graphs should be improved (Figure 4) because it is not clear, consider, normalize. 
For (521-522) persistent substances, toxicity tests are usually done. If you had them, it would be quite important because it verifies that there are no metabolites with acute toxicity.

Comments on the Quality of English Language

As mentioned above, there are some errors in the editing, improvement is needed.

Author Response

Reviewer1

Open Review

(x) I would not like to sign my review report

( ) I would like to sign my review report

Quality of English Language

(x) The English could be improved to more clearly express the research.

( ) The English is fine and does not require any improvement.

Yes

Can be improved

Must be improved

Not applicable

Does the introduction provide sufficient background and include all relevant references?

(x)

( )

( )

( )

Is the research design appropriate?

( )

( )

(x)

( )

Are the methods adequately described?

( )

(x)

( )

( )

Are the results clearly presented?

( )

( )

(x)

( )

Are the conclusions supported by the results?

(x)

( )

( )

( )

Are all figures and tables clear and well-presented?

( )

( )

(x)

( )

Comments and Suggestions for Authors

I congratulate you on your tangible and applied research. Although much research has been done on antibiotic disposal, this case highlights the application in a closed system with costing. Your research 10.3390/antibiotics12050932 is very similar to this one with the exception of the number of antibiotics studied. That is why you should focus on what is the difference and the importance of analyzing the 11 antibiotics included here, within the same category.

Response to reviewer:

We are very grateful for your insightful comments on our manuscript.

Response to reviewer:

In this study, we extended our monitoring scope to include 11 antimicrobial agents and implemented an additional intervention aimed at suppressing the proliferation of ozone-resistant environmental bacteria (e.g., Pseudomonas and Acinetobacter) that tend to adhere to the surfaces of ozonation tanks. To mitigate bacterial attachment and biofilm development, the ozonation tank was redesigned with a circular structure to promote localized water stagnation, thereby enabling the incorporation of floating hollow balls. These hollow balls were introduced to facilitate continuous mechanical cleaning by repeatedly colliding with the inner tank walls. This modification effectively minimized bacterial adhesion and contributed to maintaining a clean internal surface during long-term operation. To the best of our knowledge, this is the first report to describe the successful development and operation of a hospital wastewater treatment system capable of processing the entire volume of effluent discharged to the public sewer network, while maintaining functionality under continuous use. We believe that this approach offers a novel and practical solution to reduce the dissemination risk of AMR through hospital effluents, while simultaneously incorporating environmental sustainability into system design. The above findings and perspectives have been incorporated into the discussion section to emphasize the potential applicability and impact of this system.

I would like to make a few comments for improvement.

Sometimes the information is repeated and can become confusing (144-145 and 153). Based on this, the writing needs to be improved.

Response to reviewer:

Redundant sentence was revised as reviewer suggested.

In methodology it is mentioned that a screen is used, but no specifications are given (155).

Response to reviewer:

The description about the screen was revised as “a square stainless steel screen layer (0.12 m3) with one 10 mm mesh, one 5 mm mesh, and one 3 mm mesh, arranged in steps of increasing fineness”.

It is not clear how many samples were taken, and if it is only one, perhaps it is not significant in the day to day life of a hospital. Were repetitions made; this is due to the complexity of the sample being real wastewater.

Response to reviewer:

The description was incorrect; it has been revised as follows.

“A 100 mL aliquot of the solution was collected 0, 1, 2, 3, 4, and 5 days after the start of the experiment from each tank daily.”

Also the figures and graphics need to improve their quality. And although images help to see the big picture, graphs are more objective (Figure 2, 3), so only quality graphs are sufficient.

Response to reviewer:

Quality of figures (2, 3, and 4) are improved.

It would be interesting to do a statistical study to affirm that ozone really makes a big difference with respect to the reference. In figure 3 it seems that these two have the same behavior. And this comparison does not come in the paper.

Response to reviewer:

In line 293, although viable bacterial counts decreased significantly on day 1 post-treatment (Figure 2C), metagenomic DNA sequencing revealed that the reduction of microbial DNA occurred more gradually. This suggests that bacterial viability is rapidly lost upon initial contact with ozone molecules, but fragments of genomic DNA may persist in the solution, indicating that additional time is required for complete DNA degradation.

The graphs should be improved (Figure 4) because it is not clear, consider, normalize. 

Response to reviewer:

Quality of figures (2, 3, and 4) are improved.

For (521-522) persistent substances, toxicity tests are usually done. If you had them, it would be quite important because it verifies that there are no metabolites with acute toxicity.

Response to reviewer:

This study addresses the treatment of hospital wastewater containing a complex mixture of contaminants as raw sewage. The primary objective of this treatment is not the complete elimination of all environmental pollutants, but rather the effective reduction of residual antimicrobials. While ecotoxicological assessments are often conducted when treated effluents are directly discharged into natural water bodies, in this case, the treated hospital wastewater is subsequently introduced into a municipal wastewater treatment plant. There, it undergoes additional treatment before final discharge into the aquatic environment. As a result, the effluent remains categorized and managed as standard municipal sewage, and its potential environmental impact is mitigated through this multistage treatment process. Looking forward, it is essential to incorporate comprehensive ecotoxicity assessments of both treated hospital wastewaters and their downstream effects on wastewater treatment processes and receiving water bodies. This consideration has been incorporated into the discussion section as an important future perspective.

Comments on the Quality of English Language

As mentioned above, there are some errors in the editing, improvement is needed.

Submission Date

20 June 2025

Date of this review

01 Jul 2025 01:20:56

Reviewer 2 Report

Comments and Suggestions for Authors

This study evaluates the success of a continuous flow ozone and UV-LED-based advanced wastewater treatment system installed in a hospital in Japan. It is said to remove antimicrobial-resistant bacteria (ARB) and residual antibiotics effectively. The results show that the system inactivates more than 99% (up to 99.99%) of ARBs and completely removes some antibiotics and, to a limited extent, others.

The abstract section is enough and informative.

The introduction of the article can include a comparison of current technologies worldwide for the removal of ARB and antibiotic residues in hospital wastewater and the system used in this study. In addition, the importance of this issue in terms of environmental and public health should be emphasized in the context of legal regulations in Japan and globally. Finally, the novelty of the study can be more emphasized.

The articles below can also be added:

Region, A., Region, S. E. A., Region, E. M., & Region, W. P. (2015). Global action plan on antimicrobial resistance. Microbe Mag10(9), 354-355.

Willemsen, A., Reid, S., & Assefa, Y. (2022). A review of national action plans on antimicrobial resistance: strengths and weaknesses. Antimicrobial Resistance & Infection Control11(1), 90.

The materials and methods section and discussion parts are enough.

The limitations section is well thought out and effectively engages the readers.

Author Response

Reviewer2

Open Review

(x) I would not like to sign my review report

( ) I would like to sign my review report

Quality of English Language

( ) The English could be improved to more clearly express the research.

(x) The English is fine and does not require any improvement.

Yes

Can be improved

Must be improved

Not applicable

Does the introduction provide sufficient background and include all relevant references?

(x)

( )

( )

( )

Is the research design appropriate?

( )

(x)

( )

( )

Are the methods adequately described?

(x)

( )

( )

( )

Are the results clearly presented?

( )

(x)

( )

( )

Are the conclusions supported by the results?

(x)

( )

( )

( )

Are all figures and tables clear and well-presented?

(x)

( )

( )

( )

Comments and Suggestions for Authors

This study evaluates the success of a continuous flow ozone and UV-LED-based advanced wastewater treatment system installed in a hospital in Japan. It is said to remove antimicrobial-resistant bacteria (ARB) and residual antibiotics effectively. The results show that the system inactivates more than 99% (up to 99.99%) of ARBs and completely removes some antibiotics and, to a limited extent, others.

The abstract section is enough and informative.

The introduction of the article can include a comparison of current technologies worldwide for the removal of ARB and antibiotic residues in hospital wastewater and the system used in this study. In addition, the importance of this issue in terms of environmental and public health should be emphasized in the context of legal regulations in Japan and globally. Finally, the novelty of the study can be more emphasized.

The articles below can also be added:

Region, A., Region, S. E. A., Region, E. M., & Region, W. P. (2015). Global action plan on antimicrobial resistance. Microbe Mag, 10(9), 354-355.

Willemsen, A., Reid, S., & Assefa, Y. (2022). A review of national action plans on antimicrobial resistance: strengths and weaknesses. Antimicrobial Resistance & Infection Control, 11(1), 90.

Response to reviewer:

We revised the Introduction to include additional content and references concerning the treatment of antimicrobial-resistant bacteria and residual antimicrobials in hospital wastewater. Furthermore, we highlighted the distinctions between laboratory-scale and full-scale treatment systems to provide greater contextual clarity. Further, in the Discussion section, we expanded on the significance of the study in terms of its implications for environmental protection and public health, and incorporated relevant literature to support these points. We also emphasized the novelty of our findings and the new perspectives offered by this research in the revised manuscript.

WHO’s AMR report was cited as the first reference in the Introduction to underscore the global relevance of AMR. In addition, a second reference was incorporated to further support the contextual framework of the study.

The materials and methods section and discussion parts are enough.

The limitations section is well thought out and effectively engages the readers.

Response to reviewer:

We are very grateful for your insightful comments on our manuscript.

Submission Date

20 June 2025

Date of this review

26 Jun 2025 13:54:26

Reviewer 3 Report

Comments and Suggestions for Authors

Dear authors, 

The contamination of community wastewater and environmental waters by antibiotics and antibiotic-resistant bacteria is a growing problem worldwide.

Hospitals and other medical facilities are the main sources of this contamination.

The approach of treating hospital wastewater before it is discharged into the community is therefore promising, and it is important to investigate the technical feasibility of this approach.
Your study is well done, the results are significant, and its implications for public health are high.

In my opinion, the article is well written and could be published as is.

Thanks for your contribution!

Author Response

Reviewer3

Open Review

(x) I would not like to sign my review report

( ) I would like to sign my review report

Quality of English Language

( ) The English could be improved to more clearly express the research.

(x) The English is fine and does not require any improvement.

Yes

Can be improved

Must be improved

Not applicable

Does the introduction provide sufficient background and include all relevant references?

(x)

( )

( )

( )

Is the research design appropriate?

(x)

( )

( )

( )

Are the methods adequately described?

(x)

( )

( )

( )

Are the results clearly presented?

(x)

( )

( )

( )

Are the conclusions supported by the results?

(x)

( )

( )

( )

Are all figures and tables clear and well-presented?

(x)

( )

( )

( )

Comments and Suggestions for Authors

Dear authors,

The contamination of community wastewater and environmental waters by antibiotics and antibiotic-resistant bacteria is a growing problem worldwide.

Hospitals and other medical facilities are the main sources of this contamination.

The approach of treating hospital wastewater before it is discharged into the community is therefore promising, and it is important to investigate the technical feasibility of this approach.

Your study is well done, the results are significant, and its implications for public health are high.

In my opinion, the article is well written and could be published as is.

Thanks for your contribution!

Response to reviewer:

We are very grateful for your insightful comments on our manuscript.

Submission Date

20 June 2025

Date of this review

29 Jun 2025 13:29:57

Round 2

Reviewer 1 Report

Comments and Suggestions for Authors

Thank you for providing the requested explanations. The distinguishing factor is now clearer, although the request made could be have considered redundant. This is because your previous article is very similar in design and structure.

Best Regards

Comments on the Quality of English Language

Below are some examples of errors in the text, so that they can be avoided:
lines 81, 165-166, 155, 159.
“The wastewater was transferred from the UV treatment tank to a UV treatment unit”: no "UV treatment unit" is seen in the figure.